# Factors associated with chronic opioid use after minimally invasive lung resections

Isabel Emmerick[1]*, Hayley Reddington[1], Tanmay N. Patil[1], Alexander Neamtu[1], Jiddu Guart[1], Rebecca Foley[1], Allison Crawford[1], Karl Uy[2], Mark W. Maxfield[1], Yury Rabotnikov[2], William Phillips[1], Feiran Lou[1]

**1** University of Massachusetts Chan Medical School, Department of Surgery, Worcester, Massachusetts, United States of America, **2** University of Massachusetts Chan Medical School, Department of Anesthesia, Worcester, Massachusetts, United States of America

* isabel.emmerick@umassmed.edu

## Abstract

### Background

Individuals undergoing lung resections experience persistent postoperative pain and are at high risk of chronic postoperative opioid use. This study aims to identify factors associated with chronic opioid use after minimally invasive lung resections (MILR).

### Study Design

This is a retrospective cohort study of individuals who underwent MILR from March 2019 to May 2022 at a single academic institution. The primary outcome was chronic opioid usage, defined as use at least 30 days after surgery. Postoperative pain was managed with a standardized multi-modal pain-control regimen utilizing opioids only as needed. Prescription patterns and dispensing data of opioids at 30-, 60-, and 90-days postoperatively informed usage. Univariate analysis and multivariable logistic regressions (MVLR) were performed.

### Results

376 patients were included, 38.6% male, 88.8% white, and a mean age of 64.6 years. A total of 248 (66%) underwent anatomical lung resections. 16.5% used opioids at 30 days, 10.1% at 60 days, and 8.5% at 90 days. In the multivariable model, morphine milligram equivalents (MMEs) of opioids on the day before discharge showed a statistically significant association with chronic opioid usage. Age, sex, length of stay, and surgery type were not associated. A 10-unit increase in MMEs increased odds of use at 30-days by 21% (OR 1.21, 95%CI 1.11–1.32, p < 0.001), 20% at 60-days (OR 1.20, 95%CI 1.09 1.32, p < 0.001) and 18% at 90-days (OR 1.18, 95%CI 1.06–1.30, p = 0.002).

**Data availability statement:** Data cannot be shared publicly because the data contain potentially identifying or sensitive patient information. Data are available from the UMass Chan Medical School Institutional Data Access / Ethics Committee (contact via (508) 856-4261) for researchers who meet the criteria for access to confidential data.

**Funding:** The author(s) received no specific funding for this work.

**Competing interests:** The authors have declared that no competing interests exist.

**Abbreviations:** ASA, American Society of Anesthesiologists; ATC, Around the Clock; BMI, Body Mass Index; CAD, Coronary Artery Disease; CDC, Centers for Disease Control; CI, Confidence Interval; CPT, Current Procedural Terminology; CT, Chest Tube; CVA, Cerebrovascular Accident; DVT/PE, Deep Vein Thrombosis/Pulmonary Embolism; EMR, Electronic Medical Record; ERAS, Enhanced Recovery After Surgery; IRB, Institutional Review Board; LOS, Length of Stay MILR, Minimally Invasive Lung Resection; MI, Myocardial Infarction; MME, Morphine Milligram Equivalent; MVLR, Multivariable Logistic Regression; OR, Odds Ratio; POD, Postoperative Day; PTPS, Post-Thoracotomy Pain Syndrome; RATS, Robotic-Assisted Thoracoscopic Surgery; SEER, Surveillance, Epidemiology, and End Results; SUD, Substance Use Disorder; TIA, Transient Ischemic Attach; UMMHC, UMass Memorial Healthcare; VATS, Video-Assisted Thoracoscopic Surgery.

## Conclusion

Higher pre-discharge MMEs are associated with an increased likelihood of chronic opioid usage. Future studies should focus on whether preemptive early outpatient intercostal nerve blocks or cryoablations can decrease chronic narcotic usage in high-risk patients.

## Introduction

Patients undergoing lung resections can experience persistent postoperative pain, which is a known risk factor for chronic opioid use [1–4]. In the setting of the US opioid epidemic, which claimed over 80,000 lives via overdose in 2023, this is an incredibly relevant issue when considering patients for surgery and managing their postoperative pain [5,6]. After any surgical procedure, there are concerns about the risks of chronic opioid use, addiction, and an impact on patient quality of life. This is especially true within thoracic surgery. Thoracic surgery patients experience some of the highest rates of persistent post-operative new opioid usage compared to other surgical specialties, with early postoperative pain intensity being found to be predictive of chronic postoperative pain in this population [7]. Approximately 1 in 4 thoracic surgery patients will experience chronic post-surgical pain after a minimally invasive lung resection (MILR), emphasizing the pervasive impact of this issue within the specialty [8]. Current efforts to mitigate the detrimental effects of peri-operative opioid usage in thoracic surgery are limited by a lack of comprehensive information regarding the predictors of chronic pain [2,8–10].

Prior efforts to identify risk factors for chronic pain in thoracic surgery have been conducted with an emphasis on post-thoracotomy pain syndrome (PTPS) [11,12]. Nevertheless, as surgical practice for lung resections shifts predominantly to minimally invasive approaches, it becomes relevant to identifying risk factors associated with new persistent opioid use in this patient population [13–16]. Factors identified with an increased likelihood of chronic use after surgery include younger age, male sex, a history of substance use disorder, and cancer surgery [7]. Given the increased prevalence of video-assisted thoracoscopic surgeries (VATS) and robotic-assisted thoracoscopic surgeries (RATS), it is crucial to understand these factors in the context of MILR.

MILRs specifically have unique considerations regarding postoperative pain. There is a known neuropathic component in a subset of patients with postoperative pain, which is attributed to intercostal nerve injury or irritation [7,17,18]. Intercostal neuropraxia is specific to thoracic surgery and has been found to cause greater analgesia use and impact patients' quality of life and ability to complete daily activities [7,19]. As an approach, MILRs have been found to result in less associated postoperative pain than resections performed via thoracotomy incisions.

In other surgical specialties, [20] post-discharge opioid consumption was predictive of opioid usage on the day before discharge. The predictive value of opioid usage within 24 hours of discharge has been reiterated in the colorectal, pediatric,

and gynecologic surgical literature [21–23]. The role of inpatient use on post-discharge opioid consumption in patients undergoing thoracic surgery procedures has not yet been evaluated.

Analyzing the factors associated with chronic opioid use after a MILR is critical to inform prescription patterns, identify high-risk patients, and understand the clinical variables associated with becoming a chronic opioid user. Therefore, our study aims to identify factors associated with chronic opioid use after MILR, bridging this gap in the literature. We hypothesize that there is a positive association between the usage of opioids in the inpatient setting and the length of postoperative outpatient opioid usage.

## Materials and methods

### Study design and data source

This is a single-center retrospective cohort study of 376 patients who underwent MILR from March 2019 to May 2022 at a large academic medical center. Medical record data was accessed retrospectively for research purposes, concluding December 16, 2024. The study was approved by the University of Massachusetts Chan Institutional Review Board (IRB), study number H00017754. The need for consent was waived by the IRB, and retrospective chart review data were anonymized to prevent identification of individual participants. The goal of the study was to identify factors associated with chronic opioid use after MILR. The primary outcome was chronic opioid usage, defined as filling a prescription for opioid medication at least 30 days after surgery. Long-term opioid use was defined as having filled one or more prescriptions in the first 90 days after surgery as well as another prescription in the 90-to-180-day period following surgery.

Data was extracted from the UMass Memorial Healthcare (UMMHC) electronic medical record (EMR) system and stored in REDCap and in a secure Microsoft TEAMS folder. The demographic variables of interest include age, gender, race, and living situation. Preoperative patient characteristics include body mass index (BMI), medical comorbidities, smoking status, major psychiatric disorder, and dementia or neurocognitive dysfunction. Postoperative variables were selected to include duration of anesthesia for the patient's operation, inpatient opioid use measured as morphine milligrams equivalent (MME) postoperative days one through five, 30-day readmission rates, and post-operative complications.

### Study cohort

Patients were selected for inclusion if they were at least eighteen years of age or older, opioid-naive, and underwent a MILR. Resections included lobectomy, segmentectomy, and wedge resection based on current procedural terminology (CPT) codes. Patients were excluded from the study if they had a record of opioid medication prescribed in the 30 days before surgery (not opioid-naive). Patients who underwent resections via a thoracotomy incision, operations that were not lung resections, and operations that were converted from a minimally invasive approach were excluded.

### Institution pain management protocol

Patient postoperative pain was managed with a standardized multi-modal pain control regimen utilizing opioids only as needed, as outlined by our enhanced recovery after surgery (ERAS) pathway [24]. The standardized multi-modal pain-control regimen used consists of pre-operative administration of 975 milligrams (mg) of acetaminophen and 300 mg of gabapentin once, intra-operative administration of intercostal nerve blocks using bupivacaine 20 milliliters (mL)/1.33 percent (13.3 mg/mL), postoperative administration of around the clock (ATC) acetaminophen and gabapentin, and opioids as needed for breakthrough pain.

Patients receive discharge instructions for ATC acetaminophen, prescriptions for gabapentin 300 mg every 8 hours, and oxycodone 5 mg every 4–6 hours as needed. Opioid prescriptions upon discharge are written with a limited number of tablets. If additional opioid prescriptions were required, the patient needed to contact the thoracic surgery clinic to discuss

pain management further and refill opioid prescriptions if deemed appropriate. In case the patient's pain was not adequately managed without opioid usage four to six weeks after the surgery, then a referral to the pain management team was placed for the administration of intercostal nerve blocks.

### Statistical analysis

The cohort was stratified into groups based on opioid use at three time points: 30 days, 60 days, and 90 days post-discharge. Prescription patterns and dispensing data of opioids at 30-, 60-, and 90-days postoperatively informed usage. MME calculations were performed using standard conversion rates to determine opioid usage on a standardized scale [25]. For univariate analyses all opioid use groups were compared to one common group of "non-chronic opioid use" defined as no opioid use after 30 days. Categorical variables are expressed as number and proportion; differences across groups were tested with chi-square or Fisher's exact test. Continuous variables are expressed as means with standard deviation, with differences tested using the t-test. Separate multivariable logistic regression models were fit predicting opioid use at 30-, 60-, and 90-days after discharge, each controlling for predetermined clinically relevant variables. All analyses were conducted using SAS version 9.4.

## Results

### Patient characteristics

A total of 376 patients underwent MILR between March 2019 and May 2022 and met inclusion criteria, with 83.5% of patients having no opioid use beyond 30 days postoperatively, and 16.5% continuing opioid use at 30 days. Among these, 10.1% had opioid use at 60 days, and 8.5% at 90 days. The cohort was 39% male with a mean age of 64.6 years old and 88.8% of patients were white. The mean body mass index (BMI) for the cohort was overweight at 29.1 kg/m$^2$. There was no significant difference in gender, age, race, or BMI amongst the opioid use subgroups (Table 1).

The most common comorbidities were hypertension, coronary artery disease, diabetes, major psychiatric disorder, past or current smoking status, and atrial fibrillation. The least common were cerebrovascular accident (CVA), pulmonary hypertension, and deep vein thrombosis/pulmonary embolism (DVT/PE) (Table 1). The majority of patients in the cohort had at least one comorbid condition (64.1%) and most (66%) underwent anatomical lung resections.

### Intraoperative and postoperative variables

The mean length of anesthesia was similar across the groups, with no significant difference in length of stay (LOS) between those who used opioids beyond 30 days and those who did not (Table 2).

Within the first five postoperative days (PODs), opioid consumption was significantly associated with chronic opioid use beyond 30 days. Mean MME consumption on POD one was higher in patients with opioid use at 30, 60, and 90 days, compared to those who discontinued opioid use within 30-days following surgery (Fig 1) (Table 2). This trend continued across the first five PODs, with significant differences in MME consumption over POD two through five (Table 2).

### Chronic opioid use

Multivariable logistic regression analyses identified the MME quantity consumed on the day prior to discharge as a strong predictor of opioid use at 30, 60, and 90 days postoperatively. For each 10-unit increase in MME on the day prior to discharge, the odds of opioid use beyond 30 days increased by 21% (Fig 2) (Table 3).

Similar associations were observed at 60 days with a 20% increase and 90 days with an 18% increase (Fig 2) (Table 3). LOS per one day increase was not a significant predictor in any of the models, nor was anatomic resection. Higher MME use in the immediate postoperative period, particularly on the day prior to patient discharge, was a significant predictor of chronic and long-term opioid use in this cohort.

**Table 1. Demographic, preoperative, and operative characteristics for the full cohort and stratified by opioid use duration.**

| Variables | Total | No Opioid after 30 days | Opioid Use 30 days | | Opioid Use 60 days | | Opioid Use 90 days | |
|---|---|---|---|---|---|---|---|---|
| | n = 376 | n = 314 | n = 62 | p-value[a,b] | n = 38 | p-value[a,b] | n = 32 | p-value[a,b] |
| | mean (std) | | | | | | | |
| Age | 64.6 (10.8) | 64.8 (11.2) | 63.5 (8.4) | 0.30 | 64.2 (10.0) | 0.74 | 64.4 (9.8) | 0.85 |
| BMI | 29.1 (7.2) | 29.0 (7.1) | 29.9 (8.0) | 0.36 | 30.2 (8.2) | 0.34 | 30.5 (8.8) | 0.26 |
| | n (%) | | | | | | | |
| Male gender | 145 (38.6) | 122 (38.9) | 23 (37) | 0.77 | 19 (50) | 0.20 | 15 (46.9) | 0.39 |
| White race | 334 (88.8) | 274 (87.3) | 60 (96.8) | 0.03 | 36 (94.7) | 0.15 | 30 (93.8) | 0.24 |
| Hypertension | 208 (55.3) | 176 (56) | 32 (56.5) | 0.52 | 19 (50) | 0.48 | 17 (53.1) | 0.75 |
| CHF | 7 (1.9) | 7 (2.2) | 0 (0) | 0.61 | 0 (0) | 0.99 | 0 (0) | 0.99 |
| CAD | 65 (17.3) | 55 (17.5) | 10 (16.1) | 0.79 | 5 (13.2) | 0.50 | 5 (15.6) | 0.79 |
| MI | 13 (3.5) | 9 (2.9) | 4 (6.5) | 0.24 | 3 (7.9) | 0.13 | 3 (9.4) | 0.09 |
| Atrial fibrillation | 23 (6.1) | 18 (5.7) | 5 (8.1) | 0.56 | 3 (7.9) | 0.48 | 3 (9.4) | 0.43 |
| Valvular heart disease | 7 (1.9) | 7 (2.2) | 0 (0) | 0.61 | 0 (0) | 0.99 | 0 (0) | 0.99 |
| Pulmonary hypertension | 6 (1.6) | 5 (1.6) | 1 (1.6) | 0.99 | 0 (0) | 0.99 | 0 (0) | 0.99 |
| Major vascular disease | 17 (4.5) | 12 (3.8) | 5 (8.1) | 0.17 | 2 (5.3) | 0.65 | 2 (6.3) | 0.38 |
| Cerebrovascular history (n=347) | | | | 0.46 | | 0.14 | | 0.10 |
| CVA | 4 (1.2) | 4 (1.4) | 0 (0) | | 0 (0) | | 0 (0) | |
| TIA | 10 (2.9) | 7 (2.4) | 3 (5.2) | | 3 (8.6) | | 3 (10) | |
| None | 333 (96) | 278 (96.2) | 55 (94.8) | | 32 (91.4) | | 27 (90) | |
| DVT/PE | 6 (1.6) | 5 (1.6) | 1 (1.6) | 0.99 | 1 (2.6) | 0.50 | 0 (0) | 0.99 |
| Diabetes | 55 (14.6) | 47 (15) | 8 (12.9) | 0.67 | 7 (18.4) | 0.58 | 7 (21.9) | 0.31 |
| Current dialysis | 0 (0) | | | n/a | | n/a | | n/a |
| Coexisting cancer | 14 (3.7) | 11 (3.5) | 3 (4.8) | 0.71 | 2 (5.3) | 0.63 | 2 (6.3) | 0.33 |
| Any Comorbidity | 241 (64.1) | 199 (63.4) | 42 (67.7) | 0.51 | 27 (71.1) | 0.35 | 24 (75) | 0.25 |
| Smoking status (n=339) | | | | 0.13 | | 0.76 | | 0.34 |
| Current | 90 (26.5) | 70 (24.9) | 20 (34.5) | | 8 (23.5) | | 5 (17.2) | |
| Never | 39 (11.5) | 36 (12.8) | 3 (5.2) | | 3 (8.8) | | 2 (6.9) | |
| Past | 210 (61.9) | 175 (62.3) | 35 (60.3) | | 23 (67.6) | | 22 (75.9) | |
| Current or past smokers, mean (std) pack years | 48.7 (26.1) | 46.9 (24.4) | 54.8 (31.1) | 0.25 | 67.9 (31.6) | 0.04 | 68.8 (22.5) | 0.09 |
| Living status (n=319) | | | | 0.36 | | 0.25 | | 0.54 |
| Assisted living | 2 (0.6) | 1 (0.4) | 1 (1.7) | | 1 (2.9) | | 0 (0) | |
| Lives alone | 68 (21.3) | 57 (21.9) | 11 (18.6) | | 8 (22.9) | | 8 (27.6) | |
| Lives with family or friend | 249 (78.1) | 202 (77.7) | 47 (79.7) | | 26 (74.3) | | 21 (72.4) | |
| Major psychiatric disorder | 46 (12.2) | 37 (11.8) | 9 (14.5) | 0.84 | 8 (21.1) | 0.18 | 6 (18.8) | 0.41 |
| Dementia or neurocognitive dysfunction | 6 (1.6) | 6 (1.9) | 0 (0) | 0.60 | 0 (0) | 0.99 | 0 (0) | 0.99 |
| ASA classification (n=354) | | | | 0.56 | | 0.27 | | 0.50 |
| II | 44 (12.4) | 39 (13.3) | 5 (8.3) | | 2 (5.6) | | 2 (6.7) | |
| III | 297 (83.9) | 244 (83) | 53 (88.3) | | 34 (94.4) | | 28 (93.3) | |
| IV | 13 (3.7) | 11 (3.7) | 2 (3.3) | | 0 (0) | | 0 (0) | |
| Elective surgery | 353 (93.9) | 293 (93.3) | 60 (96.8) | 0.99 | 36 (94.7) | 0.99 | 30 (93.8) | 0.99 |
| Anatomic resection | 248 (66) | 201 (64) | 47 (75.8) | 0.07 | 26 (68.4) | 0.59 | 22 (68.8) | 0.59 |
| Digital chest tube | 56 (14.9) | 44 (14) | 12 (19.4) | 0.28 | 6 (15.8) | 0.77 | 6 (18.8) | 0.43 |

Abbreviations: std, standard deviation; BMI, body mass index; CHF, congestive heart failure; CAD, coronary artery disease; MI, myocardial infarction; CVA, cerebrovascular accident; TIA, transient ischemic attack; DVT/PT, deep vein thrombosis/pulmonary embolism; ASA, American Society of Anesthesiologists.

[a]p-value compares each group to the common control group "no opioids after 30 days."

[b]Categorical variable differences were tested with chi-square or Fisher's exact test. Continuous variable differences were tested using the t-test.

**Table 2. Postoperative outcomes and opioid use for the full cohort and stratified by opioid use duration.**

| | Total | No Opioid after 30 days | Opioid Use 30 days | | Opioid Use 60 days | | Opioid Use 90 days | |
|---|---|---|---|---|---|---|---|---|
| | n = 376 | n = 314 | n = 62 | p-value[a,b] | n = 38 | p-value[a,b] | n = 32 | p-value[a,b] |
| | mean (std) | | | | | | | |
| **Length of anesthesia (hours)** | 5.7 (1.6) | 5.7 (1.7) | 5.9 (1.5) | 0.32 | 5.8 (1.6) | 0.71 | 5.8 (1.4) | 0.58 |
| **Length of stay (days)** | 4.6 (6.8) | 4.2 (3.1) | 6.7 (15.2) | 0.21 | 7.8 (19.0) | 0.26 | 8.0 (20.7) | 0.31 |
| **Day 1 MME (n = 375)** | 65.4 (45.5) | 62.4 (45.4) | 80.8 (43.3) | 0.003 | 80.9 (45.5) | 0.02 | 84.1 (44.1) | 0.01 |
| **Day 2 MME (n = 346)** | 34.4 (36.5) | 31.0 (35.6) | 51.0 (36.7) | <0.001 | 53.7 (39.1) | <0.001 | 56.8 (40.0) | <0.001 |
| **Day 3 MME (n = 271)** | 26.7 (34.5) | 23.7 (33.0) | 41.9 (37.9) | 0.001 | 41.3 (45.5) | 0.08 | 40.3 (42.3) | 0.04 |
| **Day 4 MME (n = 173)** | 27.1 (36.7) | 22.9 (35.9) | 48.1 (33.5) | <0.001 | 50.2 (38.1) | 0.005 | 51.2 (30.3) | 0.007 |
| **Day 5 MME (n = 114)** | 24.2 (28.1) | 19.1 (23.4) | 49.9 (35.5) | 0.002 | 50.7 (38.5) | 0.01 | 46.4 (33.5) | 0.001 |
| **Mean MME over total LOS** | 42.1 (36.1) | 38.7 (35.8) | 58.8 (33.1) | <0.001 | 61.4 (36.5) | <0.001 | 63.3 (34.6) | <0.001 |
| | n (%) | | | | | | | |
| **30-day readmission** | 31 (8.2) | 25 (8.0) | 6 (9.7) | 0.72 | 5 (13.2) | 0.35 | 4 (12.5) | 0.33 |
| **Any post-op complication** | 111 (29.5) | 89 (28.3) | 22 (35.5) | 0.33 | 16 (42.1) | 0.08 | 11 (34.4) | 0.47 |
| **Unanticipated post-op invasive procedure** | 11 (2.9) | 8 (2.5) | 3 (4.8) | 0.40 | 2 (5.3) | 0.28 | 1 (3.1) | 0.58 |
| **Air leak > 5 days** | 36 (9.6) | 27 (9.0) | 9 (14.5) | 0.17 | 7 (18.4) | 0.08 | 5 (15.6) | 0.20 |
| **Pneumonia** | 10 (2.7) | 10 (3.2) | 0 (0) | 0.22 | 0 (0) | 0.61 | 0 (0) | 0.61 |
| **Pneumothorax req CT** | 8 (2.1) | 6 (1.9) | 2 (3.2) | 0.63 | 2 (5.3) | 0.21 | 0 (0) | 0.99 |
| **Atrial arrhythmia** | 12 (3.2) | 9 (2.9) | 3 (4.8) | 0.44 | 1 (2.6) | 0.99 | 1 (3.1) | 0.99 |
| **Urinary retention** | 32 (8.5) | 27 (8.6) | 5 (8.1) | 0.99 | 4 (10.5) | 0.76 | 3 (9.4) | 0.75 |

Abbreviations: std, standard deviation; MME, morphine milligram equivalents; LOS, length of stay; post-op, post-operative; req, requiring; CT, chest tube.

[a]p-value compares each group to the common control group "no opioids after 30 days."

[b]Categorical variable differences were tested with chi-square or Fisher's exact test. Continuous variable differences were tested using the t-test.

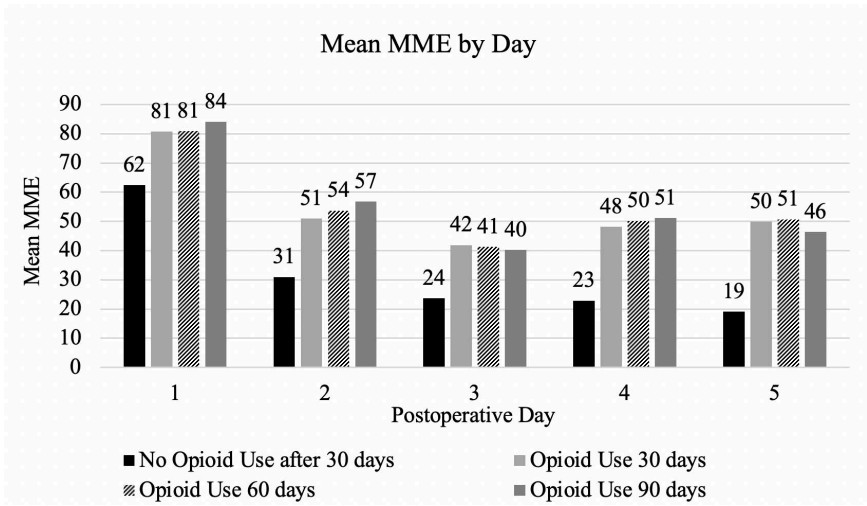

**Fig 1. Mean morphine milligram equivalents (MME) use for each subgroup by postoperative day.** Postoperative complications were not significantly different between the groups of opioid usage. Patients with opioid usage beyond 30 days had a slightly higher rate of complications, including prolonged air leak and readmission, though these were not statistically significant (Table 2). This trend in higher complication rate included any post-operative complication overall, for which the 60-day opioid use group had the highest at 42.1%. No significant difference in unanticipated postoperative invasive procedures or 30-day readmissions were observed, although these complications were in a higher proportion of patients in the groups with opioid usage beyond 30 days (Table 2).

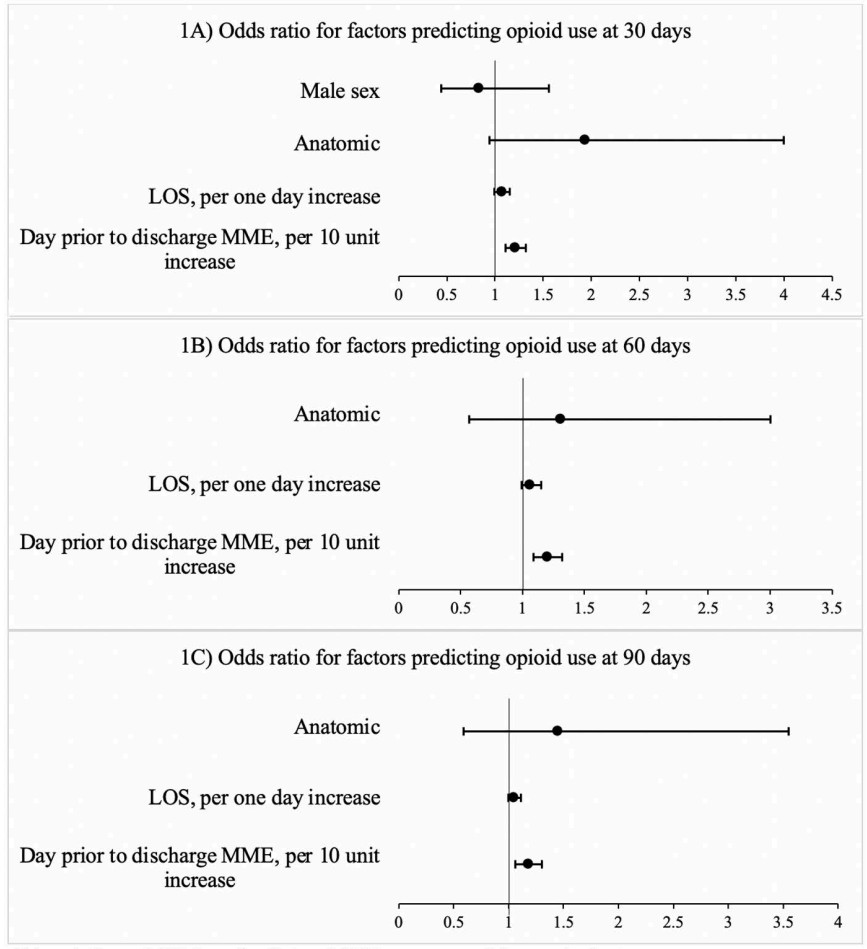

Abbreviations: LOS, length of stay; MME, mean morphine equivalents

**Fig 2. Forest plots of logistic regressions predicting opioid use at A) 30 days, B) 60 days and C) 90 days.**

## Discussion

In a retrospective analysis of postoperative opioid usage pattern after minimally invasive lung surgeries, we identified a significant association between postoperative in-hospital MME use and chronic (>30 day) opioid use, with a dose-response effect. We found that patients who consumed higher MME in the first five postoperative days, especially on the day prior to hospital discharge, were significantly more likely to continue opioid use at 30, 60, and 90 days postoperatively. The dose-response relationship is consistent with previous studies that have documented the association between perioperative opioid exposure and prolonged opioid use [1,4,10,20,26]. These findings emphasize the need for early and effective postoperative pain management. Importantly, neither LOS nor postoperative complications were significantly associated with prolonged opioid use, highlighting the central role of perioperative opioid consumption as a modifiable risk factor for chronic and long-term use. Recognition of the association is valuable in tailoring postoperative prescription guidelines.

Our findings align with prior literature that patients exposed to higher doses of opioids during the inpatient stay were more likely to continue opioid use beyond the immediate recovery period [20,21,27]. Many previous studies in thoracic surgery focused on pain after thoracotomy incisions [11,12]. Our results suggest that even in minimally invasive surgeries, where PTPS is not a concern, there is a clear association between early postoperative opioid consumption and

**Table 3. Multivariable logistic regression models fit predicting opioid use at 30-, 60-, and 90-days after discharge.**

| Logistic regression predicting opioid use at 30 days (n = 352) | OR | 95% CI | | p-value |
|---|---|---|---|---|
| Day prior to discharge MME, per 10-unit increase | 1.21 | 1.11 | 1.32 | <0.001 |
| LOS, per one day increase | 1.07 | 0.99 | 1.15 | 0.09 |
| Anatomic resection | 1.94 | 0.94 | 4.00 | 0.07 |
| Male sex | 0.83 | 0.44 | 1.56 | 0.56 |
| C statistic = 0.707 | | | | |
| **Logistic regression predicting opioid use at 60 days (n = 354)** | **OR** | **95% CI** | | **p-value** |
| Day prior to discharge MME, per 10-unit increase | 1.20 | 1.09 | 1.32 | <0.001 |
| LOS, per one day increase | 1.06 | 0.99 | 1.15 | 0.11 |
| Anatomic resection | 1.31 | 0.57 | 3.00 | 0.53 |
| C statistic = 0.708 | | | | |
| **Logistic regression predicting opioid use at 90 days (n = 354)** | **OR** | **95% CI** | | **p-value** |
| Day prior to discharge MME, per 10-unit increase | 1.18 | 1.06 | 1.30 | 0.002 |
| LOS, per one day increase | 1.05 | 1.00 | 1.11 | 0.08 |
| Anatomic resection | 1.45 | 0.59 | 3.55 | 0.42 |
| C statistic = 0.710 | | | | |

Abbreviations: OR, odds ratio; CI, confidence interval; MME, morphine milligram equivalents; LOS, length of stay.

longer-term use. Our results further highlight the need for careful management of acute postoperative pain, particularly in high-risk patients, such as those with higher immediate postoperative MME use.

Prior literature has identified associations between postoperative opioid use and younger age, male sex, lower socio-economic status, specific comorbidities (including diabetes, heart failure, and pulmonary disease), as well as preoperative history of drug abuse, alcohol abuse, depression, and benzodiazepine or antidepressant use [4,27]. In our study, MME use was the only significant predictor, with other patient characteristics and operative factors not having a significant degree of association with chronic and long-term opioid use. Anatomic resections approached but did not reach significance (P = 0.07) in the group using opioids at 30-days, however this factor became even less significant at 60-days and 90-days and was not a predictive factor in the multivariable models. Similarly, opioid use was not associated with the length of operative time. The extent of resection and time spent in the operating room were not found to be relevant to the development of chronic pain, which is likely more so related to pain pathways and patient-specific pain perception.

Further evaluation of other potential variables in predicting chronic opioid use is important to assist thoracic surgeons with preoperative planning to optimize pain management for patients. The potential predictive value of these variables has important implications as patients undergoing pulmonary resections frequently present with medical comorbidities, especially pulmonary disease, which has been hypothesized to increase sensitivity to the respiratory depressant effects of opioids, potentially increasing overdose risk [28].

The clinical implications of this study include targeting early opioid use – especially on the day prior to discharge – as a potential strategy to reduce chronic opioid use in MILR patients. Patient-centered, responsible opioid prescription guidelines have been previously evaluated in multispecialty surgical patients, including thoracic patients, based on opioid use the day prior to discharge [29]. Pre-discharge pain management protocols should be optimized to provide education to patients and minimize dependence on opioids without compromising pain control. Some interventions that have been beneficial include multimodal analgesia, preemptive early outpatient intercostal nerve blocks, and cryoablations to decrease chronic opioid usage in high-risk patients [30–32].

Given the ongoing opioid crisis in the United States, where opioid-related deaths make up the majority of overdose mortality, the implications of these findings are substantial [5,6]. Our study suggests that the current perioperative pain

management protocols, even those employing multi-modal approaches, may still contribute to the risk of chronic opioid use. Particularly troubling is that a significant proportion of the opioid-naive patients in our study had long-term use of opioids post-surgery. The association between higher in-hospital opioid consumption and chronic use suggests that further refinements in pain management may be necessary.

An important consideration that was not captured in our study is the relationship between patient-reported pain scores and postoperative opioid use. As a retrospective study, we were limited by inconsistent documentation of pain scores after discharge, which prevented their inclusion in the analysis. As a result, the study cannot determine whether higher MME usage reflected greater pain or differences in prescribing practices. Future prospective studies should incorporate standardized pain assessments to distinguish between these possibilities and clarify whether patients with higher opioid requirements are experiencing more significant postoperative pain or are at risk due to prescribing patterns alone.

As a single-center retrospective cohort study, our findings may not be broadly generalizable to other institutions with different perioperative pain management protocols. The majority of our cohort was white, which may limit the applicability to other populations. Additionally, while our multivariable logistic regression helped control for several factors, unmeasured confounders may still influence the observed associations, such as preoperative pain levels and socioeconomic factors not included as variables in the analysis. The absence of consistently recorded pain scores also limited our ability to assess whether higher opioid use was driven by patient pain versus provider prescribing behavior. Finally, reliance on prescription data to define opioid use, while a reasonable surrogate for opioid consumption, may not capture actual patient use of prescription medication [33,34].

Replicating these findings in larger, multi-center, prospective studies and examining the impact of additional perioperative factors on chronic opioid use would be valuable. Future areas of study should focus on assessing the efficacy of preemptive early outpatient interventions, such as intercostal nerve blocks and cryoablations, in reducing the need for opioids after discharge. Additionally, exploring alternative pain management strategies that lower in-hospital opioid administration could be critical in reducing chronic opioid use. The potential of enhanced recovery after surgery (ERAS) protocols in decreasing opioid use should also be investigated, as early data suggests these protocols can lead to better outcomes with lower opioid requirements [35].

## Conclusion

Immediate postoperative narcotic usage is strongly associated with chronic opioid dependence after minimally invasive lung resections. The strong association between in-hospital MMEs and continued opioid use at 30, 60, and 90-days postoperatively emphasizes the need for improved pain management protocols. Future research should continue to explore alternative analgesic strategies and their potential to reduce the risk of chronic opioid use, contributing to efforts to mitigate the opioid crisis.

## Author vcontributions

**Conceptualization:** Isabel Emmerick, Feiran Lou.

**Data curation:** Isabel Emmerick, Hayley Reddington, Tanmay N. Patil, Alexander Neamtu, Allison Crawford.

**Formal analysis:** Allison Crawford.

**Investigation:** Isabel Emmerick, Hayley Reddington, Tanmay N. Patil, Alexander Neamtu, Jiddu Guart, Rebecca Foley, Feiran Lou.

**Methodology:** Isabel Emmerick, Hayley Reddington, Allison Crawford, Feiran Lou.

**Project administration:** Isabel Emmerick, Hayley Reddington.

**Resources:** Isabel Emmerick, Hayley Reddington, Allison Crawford.

**Software:** Isabel Emmerick, Allison Crawford.

**Supervision:** Isabel Emmerick, Karl Uy, Mark W. Maxfield, Feiran Lou.

**Validation:** Isabel Emmerick, Yury Rabotnikov, William Phillips, Feiran Lou.

**Visualization:** Isabel Emmerick, Jiddu Guart, Rebecca Foley, Karl Uy, Mark W. Maxfield, Yury Rabotnikov, William Phillips, Feiran Lou.

**Writing – original draft:** Hayley Reddington, Tanmay N. Patil, Alexander Neamtu, Jiddu Guart, Rebecca Foley.

**Writing – review & editing:** Isabel Emmerick, Hayley Reddington, Tanmay N. Patil, Alexander Neamtu, Jiddu Guart, Rebecca Foley, Karl Uy, Mark W. Maxfield, Yury Rabotnikov, William Phillips, Feiran Lou.

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
