## [Decision Letter · Decision Letter 0]

1 Apr 2025

Dear Dr. Reddington,

Thank you for submitting your manuscript to PLOS ONE. After careful consideration, we feel that it has merit but does not fully meet PLOS ONE’s publication criteria as it currently stands. Therefore, we invite you to submit a revised version of the manuscript that addresses the points raised during the review process.

**ACADEMIC EDITOR:** please carefully assess all the reviewers comments

We look forward to receiving your revised manuscript.

Kind regards,

Silvia Fiorelli

Academic Editor

PLOS ONE

Additional Editor Comments (if provided):

Reviewers' comments:

Reviewer's Responses to Questions

**Comments to the Author**

1. Is the manuscript technically sound, and do the data support the conclusions?

Reviewer #1: Yes

Reviewer #2: Yes

2. Has the statistical analysis been performed appropriately and rigorously?

Reviewer #1: Yes

Reviewer #2: Yes

3. Have the authors made all data underlying the findings in their manuscript fully available?

Reviewer #1: Yes

Reviewer #2: Yes

4. Is the manuscript presented in an intelligible fashion and written in standard English?

Reviewer #1: Yes

Reviewer #2: Yes

Reviewer #1: Comments:

Title: The title clearly and concisely addresses the work done.

Abstract:

A structured abstract is presented.

Introduction

The general problem or study intent is described.

Relevant literatures were reviewed and referenced.

Methods

IRB approval is stated.

The methods are adequately explained and seem logical.

The information presented in this section is essential, concise, and appropriate.

Results

Data are presented in an understandable and logical sequence.

Reviewer #2: I think this is an interesting topic and relevant. I would be interested in the authors discussing the patients pain scores, is there a relationship between the patients pain scores and the MME of the patient. After reading the manuscript I am questioning whether these patients are in more pain requiring more narcotics or are they just being prescribed more narcotics putting them at risk of chronic opioid use. It is probably beyond the scope of the study to answer that question but I think discussion would strengthen the manuscript.

**Do you want your identity to be public for this peer review?** For information about this choice, including consent withdrawal, please see our Privacy Policy

Reviewer #1: **Yes: ** Dr Aman Edao Bime,Assistant professor of Anesthesiology

Reviewer #2: No

---

## [Author Response · Author response to Decision Letter 1]

9 Apr 2025

We thank the reviewers and the editor for their thoughtful feedback and the opportunity to revise our manuscript, “Factors associated with chronic opioid use after minimally invasive lung resections.” We have carefully addressed each comment below and made corresponding revisions to strengthen the manuscript. Our detailed responses follow:

Response to Editor:

Response: Thank you for the guidelines, we have formatted our submission to meet PLOS ONE’s style requirements.

Please confirm at this time whether or not your submission contains all raw data required to replicate the results of your study.

Response: We confirm that our submission contains all relevant data, it does not contain raw patient-level data since our institution’s protected health information is not something we can include. The Data Availability Statement has been updated to reflect this.

Response: Our 2 figures each have their own separate caption, appearing in the text at the first mention of each figure.

Response: Our reference list was reviewed, there are no citation to retracted papers and no changes were made.

Response to Reviewers:

Reviewer #1: Comments:

• Title: The title clearly and concisely addresses the work done.

• Abstract: A structured abstract is presented.

• Introduction: The general problem or study intent is described.

Relevant literatures were reviewed and referenced.

• Methods: IRB approval is stated. The methods are adequately explained and seem logical. The information presented in this section is essential, concise, and appropriate.

• Results: Data are presented in an understandable and logical sequence.

Response to Reviewer #1: Thank you for your positive and supportive comments. We appreciate the recognition of the clarity of the title, organization of the abstract, and the quality of the methodology and results presentation. No changes were made in response to this commentary.

Reviewer #2: I think this is an interesting topic and relevant. I would be interested in the authors discussing the patients pain scores, is there a relationship between the patients pain scores and the MME of the patient. After reading the manuscript I am questioning whether these patients are in more pain requiring more narcotics or are they just being prescribed more narcotics putting them at risk of chronic opioid use. It is probably beyond the scope of the study to answer that question but I think discussion would strengthen the manuscript.

Response to Reviewer #2: We appreciate your thoughtful comment and agree that the relationship between subjective pain and opioid usage is interesting and relevant. The possibility of patients receiving more pain medications due to more liberal prescription patterns should be explored in prospective studies. Unfortunately, in our analysis, routine pain scores were not reliably documented after discharge in a standardized format and therefore were not available for inclusion in the analysis. However, we have revised the discussion to address this important point, acknowledging the lack of pain score data as a limitation and discussing how this limits our ability to distinguish between pain-driven consumption versus prescription-driven risk. We have also emphasized the importance of future prospective research including pain scores to clarify this relationship.

We hope these revisions adequately address the reviewer’s concerns and improve the clarity and impact of our manuscript. We thank you again for your time and consideration

Sincerely,

Hayley Reddington, MD, on behalf of all authors

University of Massachusetts Chan Medical School

---

## [Decision Letter · Decision Letter 1]

12 May 2025

Factors associated with chronic opioid use after minimally invasive lung resections

PONE-D-25-08863R1

Dear Dr. Reddington,

We’re pleased to inform you that your manuscript has been judged scientifically suitable for publication and will be formally accepted for publication once it meets all outstanding technical requirements.

Kind regards,

Silvia Fiorelli

Academic Editor

PLOS ONE

Additional Editor Comments (optional):

Congratulations to the authors and thanks to the reviewers for the provided suggestions which really helped improve the quality of the manuscript

Reviewers' comments:

Reviewer's Responses to Questions

**Comments to the Author**

Reviewer #1: All comments have been addressed

Reviewer #2: All comments have been addressed

2. Is the manuscript technically sound, and do the data support the conclusions?

Reviewer #1: Partly

Reviewer #2: Yes

3. Has the statistical analysis been performed appropriately and rigorously?

Reviewer #1: Yes

Reviewer #2: I Don't Know

4. Have the authors made all data underlying the findings in their manuscript fully available?

Reviewer #1: Yes

Reviewer #2: Yes

5. Is the manuscript presented in an intelligible fashion and written in standard English?

Reviewer #1: Yes

Reviewer #2: Yes

Reviewer #1: I have no additional comments or concerns on this Manuscript . Every comments I made previously was well addressed

Reviewer #2: All my comments have been addressed. I think it is a well written and relevant study that addresses an important clinical concern.

**Do you want your identity to be public for this peer review?** For information about this choice, including consent withdrawal, please see our Privacy Policy

Reviewer #1: **Yes: ** Dr Aman Edao BIme,MD

Reviewer #2: No

---

## [Editor Report · Acceptance letter]

PONE-D-25-08863R1

PLOS ONE

Dear Dr. Reddington,

I'm pleased to inform you that your manuscript has been deemed suitable for publication in PLOS ONE. Congratulations! Your manuscript is now being handed over to our production team.

Kind regards,

on behalf of

Dr. Silvia Fiorelli

Academic Editor

PLOS ONE